# Deep-Learning-Based Prediction of t(11;14) in Multiple Myeloma H&E-Stained Samples

**DOI:** 10.3390/cancers17111733

**Published:** 2025-05-22

**Authors:** Nadav Kerner, Dov Hershkovitz, Svetlana Trestman, Tamir Shragai, Hila Lederman Nachmias, Yael C. Cohen, Tomer Ziv-Baran, Irit Avivi

**Affiliations:** 1Faculty of Medicine, Tel Aviv University, Tel Aviv 6997801, Israel; dovh@tlvmc.gov.il (D.H.); svetlanat@tlvmc.gov.il (S.T.); tamirsh@tlvmc.gov.il (T.S.); yaelcoh@tlvmc.gov.il (Y.C.C.); iritavi@tlvmc.gov.il (I.A.); 2Pathology Department, Tel Aviv Sourasky Medical Center, Tel Aviv 6423906, Israel; 3Hematology Division, Tel Aviv Sourasky Medical Center, Tel Aviv 6423906, Israel; 4Genetics Institute, Tel Aviv Sourasky Medical Center, Tel Aviv 6423906, Israel; hilana@tlvmc.gov.il; 5School of Public Health, Faculty of Medical & Health Sciences, Tel Aviv University, Tel Aviv 6997801, Israel; zivtome@post.tau.ac.il

**Keywords:** multiple myeloma, t(11;14), deep learning, bone marrow biopsy, detection

## Abstract

Multiple myeloma (MM) patients with t(11;14) demonstrate high response rates to BCL-2 inhibitors. Fluorescence in situ hybridization (FISH) is the gold standard for detecting t(11;14). The aim of this study was to evaluate a deep-learning-based method for detecting t(11;14) using scans of H&E-stained bone marrow biopsies of MM patients and assess factors associated with higher detection success rates. Analyzing 268 H&E bone marrow biopsies of newly diagnosed MM patients, the algorithm achieved 88% sensitivity, 83.1% specificity, 84.3% accuracy, and an area under the curve of 0.85 in cases with conclusive results. Factors associated with more advanced disease were associated with increased successful detection rates.

## 1. Introduction

Translocation of chromosomes 11 and 14 [t(11;14)(q13;32)] is the most common primary translocation in multiple myeloma (MM), occurring in approximately 20% of patients, and is associated with the overexpression of cyclin D1 and the anti-apoptotic protein B-Cell Lymphoma 2 (BCL-2) [1,2,3]. MM cells harboring t(11;14) exhibit distinct cellular features such as lymphoplasmacytic morphology, providing a rationale for attempting visual detection of such cases through histopathological analysis [1,4,5]. Moreover, both clinical prospective trials and retrospective studies have confirmed that patients with advanced relapsed/refractory MM harboring t(11;14) respond well to treatment with BCL-2 inhibitors, particularly venetoclax [6,7,8,9]. Studies exploring venetoclax-containing regimens in earlier lines of therapy have also provided very encouraging results [6,10], highlighting the need for fast and accurate detection of t(11;14) in bone marrow (BM) samples to facilitate timely treatment with BCL-2 inhibitors for eligible patients. The gold standard technique for the detection of translocations is Fluorescence in Situ Hybridization (FISH) [11,12]. However, it has limitations, including reliance on skilled technicians, specialized equipment, expensive reagents, and time-consuming processes that in clinical practice often delay cytogenetic results [13,14,15].

The use of machine learning (ML) and deep learning (DL) applications in digital pathology and molecular image-based analysis of stained slides obtained from tissue infiltrated with cancer cells can provide an immediate, objective, and scalable solution exceedingly needed [16,17,18,19]. Additionally, studies examining the use of ML tools in the diagnosis and classification of MM have shown promising results [20,21,22]. Recent studies exploring the use of DL for detecting molecular aberrations have confirmed its value across various cancers [18,23]. However, studies exploring the potential applicability and advantages of DL for cytogenetic analysis in MM are limited, and factors predicting the success of such tools in this setting are not fully elucidated [24,25,26].

The current study evaluated the performance of an AI algorithm developed by Imagene (Tel Aviv, Israel) as a rapid tool for screening BM biopsies (BMBs) for the existence of t(11;14); defining MM-related factors associated with the success of detecting this cytogenetic abnormality.

## 2. Methods

The Tel Aviv Sourasky Medical Center (TASMC) Hematology Department MM database was retrospectively reviewed, retrieving all MM patients diagnosed with active MM or smoldering MM (SMM) [27] between 2016 and 2022. Inclusion criteria were males and females, age ≥ 18 years at diagnosis, availability of good-quality diagnostic BMBs at the TASMC Pathology Department, obtained prior to anti-MM treatment, and the availability of informative FISH results analyzing at least 50 plasma cells (PCs). 

Exclusion criteria were prior exposure to an anti-MM treatment at the time of BMB sample, a known concurrent malignancy, fewer than 10% PCs in BMB, low quality of BM slides, and a low quality of FISH analysis (evaluating less than 50 PCs). Notably, cases later diagnosed with concomitant amyloidosis, extramedullary disease, or plasma cell leukemia were not excluded in order to reflect real-world workflows where BMBs are often assessed before full diagnostic data are available.

FISH was performed on BM aspirate acquired along with the BMB. FISH analysis included a two-step test for IgH translocations, starting with break-apart for IgH and, if positive, proceeding with performing FISH analysis for t(4;14) and t(11;14). In addition, 1q duplication, 1p deletion, 17p deletion, and 13q deletion were tested. The detection of t(11;14) in an analyzed sample was considered a positive case.

Hematoxylin and eosin (H&E)-stained BMB slides were scanned during 2023, using 40× magnification, and subsequently reviewed using Imagene (see below). Cases with BMB slides deemed “low-quality”, such as those that were broken or faded prior to scanning, were excluded. Demographic, pathological, and clinical data were recorded from patients’ medical notes, focusing on MM characteristics (MM subtype, staging [28], end organ involvement and MM-related blood tests [29,30], the percentage of BMB PCs, and FISH results, including the percentage of cells determined to be positive for t(11;14). 

For the development of Imagene’s algorithm, a self-supervised learning approach was employed, utilizing unannotated pan-cancer whole-slide images (WSIs) of formalin-fixed paraffin-embedded (FFPE), H&E-stained tissue samples. These images were retrieved from Imagene’s internal database, which includes slides from the TCGA Research Network and CPTAC. Subsequently, fine-tuning was performed using multiple instance learning (MIL) with WSIs from multiple myeloma (MM) patients to train the t(11;14) classifier. MIL, a form of weakly supervised learning, is particularly suited for WSI analysis, where annotations are typically available only at the slide level. A 5-fold cross-validation scheme was implemented during training. In this approach, the dataset was split into five subsets, and the algorithm was trained and validated over five iterations, each time using four subsets for training and one for validation. Validation scores from all folds were aggregated to assess overall performance. Each sample was assigned a classification of “detected”, “not detected”, or “intermediate” for t(11;14), based on prediction scores and pre-defined thresholds. The performance of the classifier was evaluated by comparing its predictions to fluorescence in situ hybridization (FISH) results. Evaluation was limited to cases with conclusive AI predictions, i.e., those classified as either “detected” or “not detected”.

To evaluate factors influencing the AI’s performance in detecting t(11;14), we measured its success rate using true positive and true negative rates. These were compared against cases classified as false positives, false negatives, or inconclusive results, all of which were considered detection failures.

Statistical analysis: Categorical variables were described as frequencies and percentages. Continuous variables distribution was evaluated using histograms and Q-Q plots, and they were reported as mean and standard deviation or as median and interquartile range or range. The chi-square test and Fisher’s exact test were applied to compare categorical variables, while the independent samples *t*-test and the Mann-Whitney test were used to compare ordinal and continuous variables. All statistical tests were two-sided, and *p* < 0.05 was considered statistically significant. Statistical analysis was performed using SPSS software (IBM SPSS Statistics for Windows, version 28, IBM Corp., Armonk, NY, USA, 2021).

## 3. Results

### 3.1. Study Population

The study investigated H&E-stained BM samples obtained from untreated, newly diagnosed MM (NDMM) patients. In total, 446 patients diagnosed with MM within the study period who had stored bone marrow biopsies (BMB) at the pathology department at TASMC and available FISH results were reviewed. A total of 178 cases were excluded from the final analysis, having non-informative FISH (n = 69), <10% PCs in BMB slides (n = 64), prior exposure to anti-MM therapy (n = 40), poor quality biopsy (n = 2), concurrent malignancy (n = 2), and lack of available scanned slides (n = 1) (Appendix A presents patient disposition).

In total, 268 cases, obtained from 147 NDMM males and 121 females, were included in the final analysis. Furthermore, 47 patients (17.5%) were diagnosed with SMM, 196 (73.1%) with active MM, 22 (8.2%) with active myeloma complicated by primary amyloidosis (AL), and 3 (1.1%) with other plasma cell dyscrasia (PCD) subtypes (Table 1). The median age of patients included in the cohort was 69.9 years (range 42–92). FISH analysis showed cytogenetic abnormalities (CAs) in 191 cases (71%) and was specifically positive for t(11;14) in 73 cases (27%). Among cases classified as t(11;14)-positive, the median percentage of identified positive cells was 26% (range 2–100%, IQR 12.5–52%). However, FISH analysis revealed fewer than 10% positive cells in 14 cases (19.2%). Additional CAs were observed in 30 cases (41%) of those positive for t(11;14), including 2 CAs in 24 cases, 3 CAs in 2 cases, and ≥4 CAs in 4 cases. No significant differences were found between patients with and without t(11;14) regarding demographics, international staging system (ISS) staging, light and heavy chain type, the percentage of PCs in BMBs, the number of cells analyzed by FISH, or laboratory characteristics (Table 1).

### 3.2. Accuracy of the AI Model

The AI determined that 102 cases (38.1%) demonstrated conclusive results regarding the expression of t(11;14). The true positive and true negative rates among conclusive cases were 21% (22/102) and 63% (64/102), respectively. There were 3 false negative (FN) cases (3%) and 13 false positive (FP) cases (13%) (Figure 1). Two of the FN cases demonstrated a low percentage of PCs (10%), lower than measured in 22 true positive cases (median 50%, IQR 20–60%). Additionally, two of the FN cases exhibited 1q gains. However, this CA was detected in 86 cases and is therefore unlikely to be the cause of misidentification. The 13 FP cases had a lower incidence of high-risk cytogenetics (specifically gain(1q) and del(17p)) than other true negative cases (*p* = 0.009), and did not differ from the true negative cases in terms of patients’ median age (69 years, IQR 50–82.8), the percentage of BM PCs (60%, IQR 10–75%), the slide age (3.7 years, IQR 2.3–5.3), or any other laboratory- and pathology-related factors (Table 2 presents univariate analysis for factors associated with false positive results).

These results translated into sensitivity and specificity among conclusive cases of 88% and 83.1%, respectively, with an accuracy rate of 84.3% and an area under the receiver operating characteristic curve (AUROC) of 0.85.

### 3.3. Factors Associated with AI Model Success 

Successful cases were defined as those in which the AI provided accurate results (n = 86). Unsuccessful cases were defined as all failures of detection, including evaluable cases with conclusive FP or FN results (n = 16) and those deemed “non-conclusive” by the AI model (n = 166). The most powerful predictor of success was the percentage of PCs in BMB (*p* < 0.001). A cutoff >40% PCs was determined to provide a 47.4% chance of success, with ≤40% providing a 23.7% chance of success (*p* = 0.001). Diagnosis of active MM (83.7% vs. 68.1%) compared with smoldering MM (10.5% vs. 20.9%) or concomitant amyloidosis (3.5% vs. 10.4%) was associated with a significantly increased success rate (*p* = 0.009). Likewise, the presence of lytic bone lesions (60.7% vs. 45.6%, *p* = 0.023) and lower hemoglobin levels (11 vs. 11.6 mg/dL, *p* = 0.025) were both associated with improved success rates (Table 3). While no specific CA was linked to the model’s success rate, the presence of additional incidental CA findings correlated with a higher success rate (15.1% vs. 6.6%, *p* = 0.025). Other pathological, demographic, prognostic, and genetic factors showed no significant correlation with the algorithm’s performance (Table 3 presents the univariate analysis of factors associated with the AI’s success).

Univariate analysis comparing conclusive versus inconclusive results identified the same factors associated with success, with conclusive cases also showing a significantly higher free light chain ratio (84 vs. 46, *p* = 0.039) (Appendix A presents this analysis).

## 4. Discussion

This study assessed the utility of a DL algorithm for detecting t(11;14) translocation in H&E-stained bone marrow biopsies of treatment-naïve MM patients. 

Our results indicate that in cases identified by the AI as having “conclusive” results, it can detect the t(11;14) translocation with high sensitivity (88%) and specificity (83.1%), achieving an overall accuracy of 84.3% and an AUROC of 0.85. These findings highlight the potential of DL tools as a rapid screening method for cytogenetic abnormalities in MM, potentially expediting personalized treatment decisions.

The gold standard for detecting t(11;14) remains FISH, which provides accurate and reliable results [11,12]. However, as discussed, FISH has limitations regarding speed, cost, and accessibility [13,14]. The deep learning approach described in our study offers a cost-effective alternative that could potentially, after further improvements, be integrated into routine clinical practice [31]. By leveraging existing H&E-stained slides, this method can circumvent the need for skilled personnel and delays in obtaining results while lowering costs in a substantial number of cases. Importantly, the algorithm’s performance metrics, particularly its sensitivity and specificity for cases determined by the algorithm as “conclusive”, approach those of FISH.

Although FISH has been well-established, our findings support the notion that AI-based methods could complement or, in certain settings, replace traditional cytogenetic testing, especially in resource-limited environments or when rapid results are required [13]. This is particularly relevant for patients with t(11;14), where timely identification can guide the use of BCL-2 inhibitors like venetoclax, which have shown superior efficacy in this specific patient population [6,7,8].

Despite the promising results, certain limitations must be addressed. The algorithm was less successful in detecting t(11;14) in cases that were characterized by more indolent disease: a lower percentage of plasma cells, a lower FLCr, and in patients with smoldering MM, reflected also by higher hemoglobin levels and the absence of lytic bone lesions. Additionally, the presence of amyloid protein appears to lower success rates. 

Surprisingly, the presence of additional cytogenetic abnormalities (CA) was shown to predict a higher success rate, probably reflecting the higher accuracy of this tool in patients with more advanced disease. Interestingly, the percentage of t(11;14)-positive plasma cells does not appear to influence the algorithm’s performance, potentially broadening its application also in cases with low translocation burden, in whom an anti-BCL-2-targeted therapy might also be useful, as suggested by the BELLINI trial, which included patients with 1% t(11;14)-positive cells [6]. Even among “conclusive” cases, the algorithm still yielded false negative and false positive results. False negative results occurred predominantly in cases with low plasma cell (PC) percentages. A plausible explanation for this finding is that, in samples with sparse PC infiltration, the morphological features indicative of t(11;14) may be less pronounced, making it difficult for the algorithm to detect the translocation. This is a significant consideration, as patients with macro-focal or patchy disease presentation might have insufficient PCs in biopsies to be accurately classified by DL models. False positive results were characterized by having a lower rate of CA associated with a higher genetic risk, specifically the absence of 1q gains and 17p deletions.

These findings emphasize the need for further refinement to enhance its accuracy in cases with low tumor burden and disease that is more indolent.

The rapid detection of t(11;14) could have profound clinical implications. Given that MM patients with t(11;14) demonstrate a favorable response to BCL-2 inhibitors, identifying these patients in early stages of the disease could facilitate timely initiation of targeted therapies, potentially improving outcomes. 

A study by Stanford-Moore et al. investigated the application of a machine learning model utilizing 218 H&E-stained slides and 213 IHC-stained slides to predict the presence of t(11;14) in multiple myeloma (MM). The study reported a sensitivity of 0.67 for the H&E model and 0.75 for the combined H&E and IHC model, achieving a combined negative predictive value of 0.96 [24]. 

Another study by Lewis et al. evaluated a machine learning tool designed to detect plasma cell neoplasms and predict multiple genetic subtypes based on Wright-stained bone marrow aspirate smears. This model identified t(11;14) with an AUROC of 0.73 [25]. A second study by this group assessed a machine learning tool that integrated data from Wright-stained bone marrow slides and myeloma flow cytometry panels to predict the presence of genetic variants in 85 myeloma cases. This study achieved an AUROC of 0.892 for detecting the presence of t(11;14) [26]. Although these studies, including ours, utilized different pathological materials, they all support the effectiveness of detecting the 11;14 translocation using an image processing AI method. Some methods demonstrated improved performance when combined with additional data sources.

These studies, along with the current study, by validating an AI-based approach, offer a step forward in achieving this goal. With further validation, such tools could be incorporated into the diagnostic workflow for MM, enabling oncologists to rapidly stratify patients and make more informed treatment decisions. In healthcare settings where FISH is not readily available, AI-based methods may provide a viable alternative, particularly in underserved regions. Moreover, as digital pathology and telemedicine become more widespread, the scalability of AI-based diagnostics may allow for broader application of precision medicine principles in MM, including in predicting responses to specific therapies [32].

Several limitations of the study design must be acknowledged. First, the retrospective nature of the study limits the generalizability of the findings. Additionally, our study focused exclusively on BMB obtained from newly diagnosed patients, whereas treatment with BCL-2 inhibitors is usually considered in patients with relapsed disease [6,7]. Moreover, as the algorithm was trained on a mixture of pan-cancer images and fine-tuned on MM-specific samples, further refinement specific to MM morphology may enhance its detection accuracy.

Future studies should adopt a multi-center, multi-phase design to address current limitations and enhance the generalizability of the AI algorithm. As a first step, a multi-institutional study should incorporate slides from a diverse range of pathology departments, using standardized processing protocols to minimize inter-laboratory variability. The patient cohort should be expanded to include both newly diagnosed and relapsed/refractory multiple myeloma (MM) cases, allowing for a comprehensive assessment of algorithm performance across different disease stages. Importantly, cases of AL amyloidosis should be excluded, as this study suggests they may reduce the algorithm’s sensitivity. As clinical outcomes data become available, a correlation analysis between algorithm-detected t(11;14) and response to venetoclax treatment should be conducted, with the potential to establish the AI model as a predictive biomarker for therapy selection.

To increase the rate of conclusive results, the algorithm should be further trained on a substantially larger dataset that includes samples with a broad range of plasma cell percentages and morphological variations.

## 5. Conclusions

In conclusion, this study demonstrates the potential of a deep learning-based approach to detect t(11;14) translocation in newly diagnosed MM patients and highlights several aspects affecting its performance. The high sensitivity and specificity observed especially in patients with features of more advanced disease suggests that this method could serve as a rapid, cost-effective supplement and potential alternative to traditional cytogenetic techniques like FISH. As BCL-2 inhibitors become more integrated into frontline therapies for MM, particularly for patients with t(11;14), the timely identification of this subgroup is critical. Further improvement, increasing rates of conclusive cases, and reducing the probability of FP results are warranted to confirm these findings and explore the full potential of AI-based diagnostics in this particular setting.

## Figures and Tables

**Figure 1 cancers-17-01733-f001:**
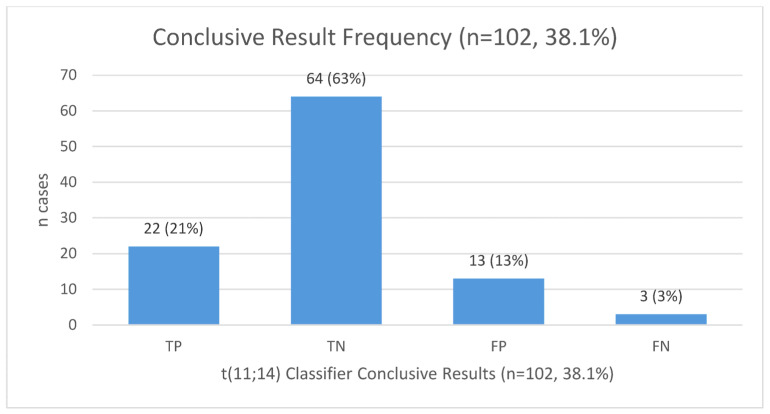
AI Classifier vs. FISH results in detecting t(11;14) in bone marrow biopsy slides. Legend: TN—true negative; TP—true positive; FN—false negative; FP—false positive.

**Table 1 cancers-17-01733-t001:** Patient characteristics.

	Entire Cohort (n = 268)	t(11;14) Positive Cohort (n = 73)	None-t(11;14) Cohort (n = 195)	*p* Value
Age years, median (range)	69.9 (41.8–92.1)	68.8 (45.7–92.1)	70.2 (41.8–91.4)	0.915
Sex, (Males), n (%)	147 (54.9)	42 (57.5)	105 (53.8)	0.589
PCD type	-			0.349
Active MM	196 (73.1)	50 (68.5)	146 (74.9)	
Smoldering MM	47 (17.5)	14 (19.2)	33 (16.9)	
Active MM with AL	22 (8.2)	7 (9.6)	15 (7.7)	
Other *	3 (1.1)	2 (2.7)	1 (0.5)	
Hypercalcemia ^‡^, n (%)	17 (7)	4 (6.3)	13 (7.2)	0.815
Renal insufficiency ^α^, n (%)	29 (11.4)	6 (8.8)	23 (12.4)	0.432
Hemoglobin mg/dL, mean (SD)	11.4 (2)	11.4 (1.84)	11.4 (2.07)	0.815
Anemia ^β^, n (%)	61 (23.7)	13 (18.6)	48 (25.7)	0.234
Lytic bone lesions, n (%)	128 (50.6)	31 (45.6)	97 (52.4)	0.334
Heavy Chain Subtype, n (%)	-			0.076
IgG	126 (48.8)	28 (40)	98 (52.4)	
Non IgG	131 (51.2)	42 (60)	89 (47.6)	
Light Chain Subtype, n (%)	-			0.129
Kappa	173 (65)	43 (58.9)	130 (67.4)	
Lambda	92 (34.6)	29 (39.7)	63 (32.6)	
Nonsecretory MM	1 (0.4)	1 (1.4)	0 (0)	
FLCr, median (IQR)	55 (8.4–214)	55 (8.4–186)	59.4 (8.5–285.8)	0.614
High risk cytogenetics ^γ^, n (%)	109 (40.7)	22 (30.1)	87 (44.6)	**0.032**
ISS	-			0.794
I	77 (42.3)	21 (42.9)	56 (42.1)	
II	45 (24.7)	13 (26.5)	32 (24.1)	
III	60 (33)	15 (30.6)	45 (33.6)	
Missing	86	24	62	
% of PCs in BMB, median (range; IQR)	30 (10–100; 20–60)	30 (10–100; 20–60)	30 (10–100; 20–50)	0.307
% of t(11;14) positive cells ^†^, median (range; IQR)	-	26 (2–100; 12.5–52)	-	
Time from BMB to screening, years, median (IQR)	3.1 (1.9–4.3)	3.1 (1.6–4.1)	3 (2–4.5)	0.735

* Includes plasma cell leukemia and solitary plasmacytoma. ^†^ Determined by FISH for patients with t(11;14) only ^‡^ Serum calcium > 11 mg/dL ^α^ Serum creatinine > 2 mg/dL ^β^ Hemoglobin of <10 g/dL ^γ^ High-risk cytogenetics was determined as positive for t(4;14), t(14;16), t(14;20), del(17p), and gain 1q. AL—Light Chain Amyloidosis; BMB—Bone marrow biopsy; FLCr—Free Light Chain Ratio; ISS—International Staging System; IQR—Interquartile Range; MM—Multiple Myeloma; PCD—Plasma cell dyscrasia; SD—Standard Deviation.

**Table 2 cancers-17-01733-t002:** Univariate analysis of factors associated with false positive results compared with true negative results.

	False Positive n = 13 (16.9%)	True Negative n = 64 (83.1%)	*p* Value
Age years, median (range)	70.5 (50–82.8)	70.1 (41.8–89.7)	0.999
Sex, (Males), n (%)	6 (46.2)	34 (53.1)	0.646
PCD type, n (%)			0.803
Active MM	11 (84.6)	54 (84.4)	
Smoldering MM	2 (15.4)	7 (10.9)	
Active MM with AL	0 (0)	2 (3.1)	
Other *	0 (0)	1 (1.6)	
Slide Age, years, median (IQR)	3.7 (2.3–5.3)	3.1 (2.3–5.2)	0.838
Number of CA, median (IQR)	0 (0–1)	1 (0–2)	0.183
High Cytogenetic Risk, n (%)	1 (7.7)	30 (46.9)	**0.009**
ISS, n (%)			0.514
I	4 (40)	13 (30.2)	
II	3 (30)	13 (30.2)	
III	3 (30)	17 (39.5)	
Missing	3	21	
% of PCs in BMB, median (range; IQR)	60 (10–90; 10–75)	50 (10–100; 20–67.5)	0.984
Calcium mg/dL, median (IQR)	9.6 (9.4–10.4)	9.4 (9–9.9)	0.123
Hypercalcemia ^‡^, n (%)	0 (0)	4 (6.7)	0.999
Renal insufficiency ^α^, n (%)	2 (16.7)	7 (10.9)	0.627
Hemoglobin mg/dL, mean (SD)	10.7 (1.6)	10.9 (2.1)	0.693
Anemia ^β^, n (%)	5 (41.7)	20 (31.3)	0.515
Lytic bone lesions, n (%)	9 (69.2)	39 (61.9)	0.757
Heavy Chain Subtype, n (%)			0.201
IgG	5 (38.5)	37 (57.8)	
Non-IgG	8 (61.5)	27 (42.2)	
Light Chain Subtype, n (%)			0.269
Kappa	10 (90.9)	45 (70.3)	
Lambda	1 (9.1)	19 (29.7)	
FLCr, median (IQR)	69 (7–488)	74 (10–262)	0.87

* Solitary plasmacytoma ^‡^ Serum calcium > 11 mg/dL ^α^ Serum creatinine > 2 mg/dL ^β^ Hemoglobin of <10 g/dL AL—Light Chain Amyloidosis; BMB—Bone marrow biopsy; CA—Cytogenetic Abnormalities; FLCr—Free Light Chain Ratio; ISS—International Staging System; IQR—Interquartile Range; MM—Multiple Myeloma; PC—Plasma cell dyscrasia; SD—Standard Deviation.

**Table 3 cancers-17-01733-t003:** Univariate analysis of factors associated with AI success, including all analyzed cases.

	Entire Cohort (n = 268)	Successful Detection of t(11;14) (n = 86, 32.1%)	Failure of Detection of t(11;14) (n = 182, 67.9%)	*p* Value
t(11;14) positive, n (%)	73 (27.2)	22 (25.6)	51 (28)	0.675
Age years, median (range)	69.9 (41.8–92.1)	69.9 (41.8–89.7)	69.8 (44.3–92.1)	0.276
Sex, (Males), n (%)	147 (54.9)	48 (55.8)	99 (54.4)	0.828
PCD type, n (%)	-			**0.009**
Active MM	196 (73.1)	72 (83.7)	124 (68.1)	
Smoldering MM	47 (17.5)	9 (10.5)	38 (20.9)	
Active MM with AL	22 (8.2)	3 (3.5)	19 (10.4)	
Other *	3 (1.1)	2 (2.3)	1 (0.5)	
% of PCs in BMB, median (range; IQR)	30 (10–100; 20–60)	50 (10–100; 20–63)	25 (10–100; 19–50)	**0.0003**
% of t(11;14) positive cells ^†^, median (range; IQR)	26 (2–100; 12.5–52)	30.5 (2–90; 15–57)	24 (3–100; 12–50)	0.358
Calcium mg/dL (median, IQR)	9.4 (9–9.9)	9.4 (8.9–9.8)	9.4 (9–9.9)	0.816
Hypercalcemia ^‡^, n (%)	17 (7)	5 (6.3)	12 (7.4)	0.749
Creatinine mg/dL, median (IQR)	0.99 (0.79–1.38)	1.02 (0.8–1.47)	0.98 (0.77–1.3)	0.444
Renal insufficiency ^α^, n (%)	29 (11.4)	10 (11.8)	19 (11.2)	0.902
Hemoglobin mg/dL, mean (SD)	11.4 (2)	11 (2.2)	11.6 (1.88)	0.025
Anemia ^β^, n (%)	61 (23.7)	24 (28.2)	37 (21.5)	0.233
Lytic bone lesions, n (%)	128 (50.6)	51 (60.7)	77 (45.6)	**0.023**
Heavy Chain Subtype, n (%)	-			0.123
IgG	126 (49)	48 (55.8)	78 (45.6)	
Non-IgG	131 (51)	38 (44.2)	93 (54.4)	
Light Chain Subtype, n (%)	-			0.783
Kappa	173 (65)	58 (68.4)	115 (63.9)	
Lambda	92 (34.6)	28 (32.6)	64 (35.6)	
Nonsecretory MM	1 (0.4)	0 (0)	1 (0.6)	
FLCr, median (IQR)	55 (8.4–214)	82 (9.4–256)	50 (8.3–182)	0.09
Number of CA, median (IQR)	1 (0–2)	1 (0–2)	1 (0–2)	0.547
Any Other CA, n (%)	25 (9.3)	13 (15.1)	12 (6.6)	**0.025**
High Cytogenetic Risk ^γ^, n (%)	109 (40.7)	38 (44.2)	71 (39)	0.421
ISS, n (%)	-			0.504
I	77 (42.3)	24 (39.3)	53 (43.8)	
II	45 (24.7)	15 (24.6)	30 (24.8)	
III	60 (33)	22 (36.1)	38 (31.4)	
Missing, n	86	25	61	
Slide Age, years, median (IQR)	3.1 (1.9–4.3)	3.1 (2.2–4.3)	3 (1.8–4.2)	0.347

* Includes plasma cell leukemia and solitary plasmacytoma ^†^ Calculated only for patients FISH positive for t(11;14) ^‡^ Serum calcium > 11 mg/dL ^α^ Serum creatinine > 2 mg/dL ^β^ Hemoglobin of <10 g/dL ^γ^ High-risk cytogenetics was determined as positive for t(4;14), t(14;16), t(14;20), del(17p) and gain 1q. AL—Light Chain Amyloidosis; BMB—Bone marrow biopsy; CA—Cytogenetic Abnormalities; FLCr—Free Light Chain Ratio; ISS—International Staging System; IQR—Interquartile Range; MM—Multiple Myeloma; PC—Plasma cell dyscrasia; SD—Standard Deviation.

## Data Availability

The data generated in this study are available upon request from the corresponding author in accordance with our institution’s safety policy for data transfer.

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
