# Peer review of "Deep-Learning-Based Prediction of t(11;14) in Multiple Myeloma H&E-Stained Samples"

_cancers, 2025, doi:10.3390/cancers17111733_

Round 1

Reviewer 1 Report

Comments and Suggestions for Authors

The manuscript entitled “Deep Learning-Based Prediction of t(11;14) in Multiple  Myeloma H&E-Stained Samples” investigates a deep learning-based method for detecting multiple myeloma based on the detection of translocation of chromosomes 11 and 14 t(11;14). It is a retrospective study on the newly diagnosed patient. This research design has many limitations and needs further studies for improvement. There are some points that should be considered to improve the final quality of this article as follows:

  • The simple summary contains the same information that already exists in the abstract. Remove the summary part to avoid repetition.
  • Some abbreviations need to be explained, like BCL-2 inhibitor = a selective inhibitor of the anti-apoptotic protein B-cell lymphoma. Full names are written the first time, and then short names can be written after this.
  • Methodology: The Imagene’s algorithm development process needs more details. For example, a 5-fold cross-validation scheme. Rephrase this part and add more details
  • Discussion: There are many limitations in this study that put constraints on the ability to generalize or apply. As a plan for improving this tool, write a short strategy describing how to overcome these limitations.
  • The format of the references is incorrect. Refer back to the author's instructions and make sure you are using the right format.

Author Response

  • Comment 1: The simple summary contains the same information that already exists in the abstract. Remove the summary part to avoid repetition.
  • Answer: We thank the reviewer for his comment. We prepared an abstract + short summary, according to the journal’s instructions. However, we have revised our summary and shortened it and would be happy to delete it if required.

  • Comment 2: Some abbreviations need to be explained, like BCL-2 inhibitor = a selective inhibitor of the anti-apoptotic protein B-cell lymphoma. Full names are written the first time, and then short names can be written after this.
  • Answer: We thank the reviewer for pointing this out. We corrected it for all abbreviations (BCL-2- explained now (line 44); TASMC- Tel Aviv Sourasky Medical Center (line 72); ISS- International Staging System (line 147); FFPE - formalin fixed paraffin embedded (line 97).

  • Comment 3: Methodology: The Imagene’s algorithm development process needs more details. For example, a 5-fold cross-validation scheme. Rephrase this part and add more details.
  • Answer: We thank the reviewer for his valuable comment. We revised this section and added details and clarifications in the Method section. In particular, the 5-fold cross-validation scheme was clearly explained and thoroughly elaborated (methods section line 96).

  • Comment 4: Discussion: There are many limitations in this study that put constraints on the ability to generalize or apply. As a plan for improving this tool, write a short strategy describing how to overcome these limitations.
  • Answer: We thank the reviewer for his important comment. We completely agree with the reviewer that this new technology is still far from being implanted in the clinic, having a high proportion of inconclusive cases, and even more importantly, insufficient specificity. We expanded the section that addresses these limitations and addressed technological and clinical strategies to mitigate this these obstacles in the discussion section ( marked in the Discussion section line 290).

  • Comment5: the format of the references is incorrect. Refer back to the author's instructions and make sure you are using the right format.
  • Answer: We thank the reviewer for his comment. We have corrected the references according to the requested MDPI format.

Reviewer 2 Report

Comments and Suggestions for Authors

This is a very interesting study. I have several comments:

1 Did the authors try, considering the number of patients, to divide the patients into training and validation cohorts? Perhaps that would increase sensitivity and specificity of the model.

2. I think it would make sense to disregard samples of MM+AL, plasma cell leukemia and the solitary plasmocytoma sample before the analysis. It should be strictly MM samples.

3. I wonder if exclusion of younger patients (around 40 and 50) would make a difference...as MM seems to be more aggressive in younger patients....

Author Response

  • Comment 1: Did the authors try, considering the number of patients, to divide the patients into training and validation cohorts? Perhaps that would increase sensitivity and specificity of the model.
  • Answer: We thank the reviewer for the insightful comment. We acknowledge that the initial description of the algorithm development was insufficient and have therefore expanded it accordingly. In response to your comment, the term 5-fold cross-validation scheme refers to the process of dividing the patient cohort into five groups and applying the algorithm five times. In each iteration, four groups were used for training while the remaining group served as the validation cohort. This approach allowed us to apply the algorithm multiple times across the same dataset, helping to mitigate the limitation of the relatively small cohort size. We agree that a crucial next step would be to validate the algorithm on a larger, independent test set. The paragraph that describes the algorithm development in the article was expanded (Methods section line 96).

  • Comment 2: I think it would make sense to disregard samples of MM+AL, plasma cell leukemia and the solitary plasmocytoma sample before the analysis. It should be strictly MM samples. We thank the reviewer for his valuable comment .
  • Answer: We believe that the current analysis—which includes patients with multiple myeloma (MM), with or without plasmacytoma, amyloidosis, and/or plasma cell involvement in the blood—is important, as it reflects the real-world clinical workflow. In practice, bone marrow biopsies are often submitted for analysis prior to the availability of additional diagnostic data, such as plasma cell (PC) staining or AL staining. Therefore, our intention was to include in the analysis all slides that met the inclusion and exclusion criteria, regardless of the final diagnosis of plasma cell dyscrasia that was ultimately assigned to those slides. This issue was adressed in the methods sections (Line 80).

  • Comment 3: I wonder if exclusion of younger patients (around 40 and 50) would make a difference...as MM seems to be more aggressive in younger patients....
  • Answer: We thank the reviewer for this comment. The algorithm was designed to identify t(11;14) rather than to predict disease aggressiveness or multiple myeloma (MM)-related risk. Furthermore, our final analysis included only 268 patients, which limits the statistical power to reliably assess the algorithm’s performance across different age groups. In addition, univariate analysis did not demonstrate any significant impact of age on the algorithm's performance. It is also important to note that the role of age in MM aggressiveness remains controversial, as suggested by previous studies (Ludwig et al., PMID: 18268097; Jurczyszyn et al., PMID: 27682187).Given these considerations, we elected to conduct the analysis across the entire cohort.

Round 2

Reviewer 1 Report

Comments and Suggestions for Authors

The authors addressed all the revision points, and I approve the manuscript for publication.